# The Binding of Phosphorus Species at Goethite: A Joint Experimental and Theoretical Study

Prasanth B. Ganta [1], Mohsen Morshedizad [2], Oliver Kühn [1,3], Peter Leinweber [2,3] and Ashour A. Ahmed [1,3,*]

[1] Institute of Physics, University of Rostock, Albert-Einstein-Str. 23–24, D-18059 Rostock, Germany; prasanth.ganta@uni-rostock.de (P.B.G.); oliver.kuehn@uni-rostock.de (O.K.)
[2] Chair of Soil Science, University of Rostock, Justus-von-Liebig-Weg 6, D-18059 Rostock, Germany; mohsen.morshedizad3@uni-rostock.de (M.M.); peter.leinweber@uni-rostock.de (P.L.)
[3] Department of Life, Light, and Matter (LLM), University of Rostock, Albert-Einstein-Str. 25, D-18059 Rostock, Germany
* Correspondence: ashour.ahmed@uni-rostock.de; Tel.: +49-381-498-6943

**Abstract:** Knowledge of the interaction between inorganic and organic phosphates with soil minerals is vital for improving soil P-fertility. To achieve an in-depth understanding, we combined adsorption experiments and hybrid ab initio molecular dynamics simulations to analyze the adsorption of common phosphates, i.e., orthophosphate (OP), glycerolphosphate (GP) and inositolhexaphosphate (IHP), onto the 100 surface plane of goethite. Experimental adsorption data per mol P-molecule basis fitted to the Freundlich model show the adsorption strength increases in the order GP < OP < IHP, and IHP adsorption being saturated faster followed by GP and OP. Modeling results show that OP and GP form stable monodentate (**M**) and binuclear bidentate (**B**) motifs, with **B** being more stable than **M**, whereas IHP forms stable **M** and **3M** motifs. Interfacial water plays an important role through hydrogen bonds and proton transfers with OP/GP/IHP and goethite. It also controls the binding motifs of phosphates with goethite. Combining both experimental and modeling results, we propose that the **B** motif dominates for OP, whereas GP forms **M** and IHP forms a combination of **M** and **3M** motifs. The joint approach plausibly explains why IHP is the predominant organically bound P form in soil. This study could be considered as a preliminary step for further studies for understanding the mechanisms of how microbes and plants overcome strong IHP–mineral binding to implement the phosphate groups into their metabolism.

**Keywords:** P-fertility; orthophosphate; glycerolphosphate; inositolhexaphosphate; goethite; adsorption; surface complexation; MD simulations; QMMM

## 1. Introduction

The phosphorus (P) adsorption capacity of soil is a crucial factor affecting the P-immobilization process and thus soil fertility, and fate of P in natural environments [1–4]. This adsorption capacity, defined as the amount of adsorbed substance (adsorbate), reached in a saturated solution for a specific adsorbent [5], is fundamentally influenced by the diverse interactions of P with soil constituents, involving free metal ions [6,7], soil minerals [8–11], and soil organic matter [12,13]. In particular, the strong interaction between phosphates and soil mineral surfaces, and especially Fe- and Al-(oxyhydr)oxides, plays a very important role in controlling this process [8,14–18]. The Fe-oxides are widespread in surface environments and constitute a major component of highly weathered soils and sediments. They have attracted considerable attention due to their high P-adsorption capacity [19–21]. Phosphate adsorption on iron oxide surfaces shows a biphasic behavior consisting of a rapid and strong ligand exchange step followed by a slower step [22–25]. The latter phase was described previously by the formation of monodentate (**M**) complexes and conversion into bidentate (**B**) complexes, the competition with other anions, and/or

precipitation events [26]. Alternatively, the slower phase has been assigned to a diffusion process of phosphate ions from an outer-sphere complex to the surface [16,23,27,28].

The extent of P-uptake during these processes depends on reactive surface groups and the degree of crystallinity or porosity of iron oxides [27,29]. In soils, sediments and natural environments, goethite is the most frequent and widespread form of iron oxy-hydroxides [30]. It has high crystallinity with a specific surface area varying from 18 to 132 $m^2\,g^{-1}$ [27,31] depending on synthesis conditions which may thus influence the P-adsorption capacity of minerals [32]. Goethite has been studied extensively since it is considered as one of the most reactive surfaces for P compounds in the environment [33]. However, conflicting discussions are still present about the nature of binding motifs of the adsorbed phosphate species and the relative abundance of **M** complexes [34,35] versus **B** ones [36,37]. This could be due to the presence of different surface planes for goethite and several binding motifs for the goethite–phosphate complexes in addition to the different setups and conditions of adsorption experiments reported in literature. Consequently, different opinions have been put forward to assign the goethite–phosphate binding motifs based on infrared (IR) spectroscopy. This involves assignment either by the positions of P–O vibrational bands [36], the number of P–O bands and not their exact position [15], or formation of only an **M** motif with different protonation states depending on pH [38]. Molecular modeling is considered as a powerful tool to resolve such conflicting discussions by providing a molecular level description for the nature of the formed goethite–phosphate complexes.

By using molecular modeling, Kwon and Kubicki [39] studied the binding process of phosphate at goethite and suggested the abundance of the diprotonated **B** complex at pH 4–6 and the deprotonated **B** or monoprotonated **M** complex at pH 7.5–7.9. In a further study [40], they predicted the formation of **M** complexes at the 001, 210, 101, and 100 surface planes and **B** complexes at the 010, 101, and 100 surface planes. Recently, we have simulated the binding mechanism of phosphate at the goethite water interface at about pH 6, considering two different goethite surface planes [10]. The outcome of this study pointed to the abundance of the monoprotonated **B** motif at the goethite–water interface and the importance of water in controlling this binding process via promoting of specific binding motifs, formation of H-bonds (HBs), adsorption and dissociation at the surface, and proton transfer processes. Furthermore, the assignment of calculated IR spectra in this study introduced a new approach for characterizing experimental IR data of adsorbed phosphate species. Investigations at a molecular level for the binding of other P-compounds to soil reactive surfaces have also been reported. For example, the mechanism, nature, strength, and different possible binding motifs of interaction of glyphosate, the most used herbicide, with goethite, as well as with representative models for soil organic matter, have been explored [9]. Moreover, we studied the interaction of organic phosphate such as glycerolphosphate (GP) and inositolphosphate (IHP) at the diaspore($\alpha$-AlOOH)–water interface by focusing on two diaspore surface planes [11,41]. The results revealed stronger interactions for both GP and IHP at the 010 diaspore surface plane compared to the 100 surface plane. Further, IHP binds stronger to both surfaces via three phosphate groups compared to GP.

Such a detailed atomistic investigation for the binding mechanisms of both inorganic and organic phosphates at the goethite–water interface is still lacking. Therefore, our objective in the present study is to introduce a molecular level understanding for the binding mechanism of phosphate at the goethite–water interface via a joint experimental/theoretical approach. Specifically, adsorption experiments have been performed for orthophosphate (OP), GP, and IHP on the pure synthetic goethite surface at constant pH and ionic strength. Further, the simulation of these adsorption experiments have been carried out by molecular modeling for the binding processes of OP/GP/IHP at the goethite–water interface, applying hybrid quantum mechanics/molecular mechanics (QM/MM)-based molecular dynamics (MD) simulations.

## 2. Materials and Methods

### 2.1. Goethite Preparation and Adsorption Experiments

Goethite was prepared as described by Dultz et al. [42], i.e., a 10 M NaOH solution was added to a 0.5 M FeCl$_3$ solution (FeCl$_3 \cdot 6$H$_2$O, Merck AG) with stirring continuously to bring the pH to 12. The resulting ferrihydrite was aged for 120 h at 55 °C to be converted to goethite. Then, the suspension was subjected to pH adjustment to pH 6 by adding 0.1 M HCl and followed by centrifugation-washing cycles with distilled water until the electrical conductivity was lower than 10 μS cm$^{-1}$. The so-prepared goethite was freeze-dried and stored as powder for further analysis and experiments.

The results of X-ray diffraction (XRD) analysis confirmed the well-crystallinity of goethite samples; see Figure S1. The measured specific surface area (SAA) of goethite by Brunauer-Emmett-Teller (BET, N$_2$ adsorption; Nova 4000e, Quantachrome, Boynton Beach, FL, USA) method is 64.5 m$^2$ g$^{-1}$. This value lies in the reported common range values of SSA of goethite in the literature [27,33,43]. It should be noted that different SSA values of goethite were reported in the literature. This could be due to the sensitivity of the BET method to the type of adsorbent, lack of an appropriate linearity criterion (a linear region within the standard pressure range assuming that monolayer adsorption will occur in this pressure range), and presence of micro pores [44]. The mean hydrodynamic diameter of goethite particles is 1.6 ± 0.2 μm as determined by a particle sizer (Zetapals, Brookhaven, Holtsville, NY, USA) on a 100 mgL$^{-1}$ suspension treated with a dispersant agent (0.01 mM Na$_4$P$_2$O$_7$, pH 8) and sonicated for 30 s (Labsonic M, Sartorius Stedim, Göttingen, Germany). The measured point of zero charge (PZC) of goethite particles is 6.3; see Figure S2.

In the present study, the adsorption of three different phosphate compounds involving inorganic orthophosphate (OP, potassium dihydrogen phosphate, CAS number: 7778-77-0) and organic phosphates (α-glycerol phosphate (GP, CAS number: 17603-42-8) and myo-inositol hexakisphosphate (IHP, CAS number: 83-86-3)) on goethite was performed. Here, a stock solution of 10 mM P of each P compound was prepared in 0.01 M CaCl$_2$ solution. For each experiment, 200 mg sample of goethite was weighed into a 50 mL centrifuge tube and equilibrated with 40 mL of each initial P concentration. Here, twelve different initial P concentrations (0, 0.05, 0.1, 0.15, 0.2, 0.3, 0.5, 1, 2, 3, 5, and 10 mM P) were considered. Each initial concentration was prepared by the dilution of the P stock in 0.01 M CaCl$_2$ background electrolyte solution, adjusted at pH 5. After an equilibrium period of 24 h at 25 °C under end-over-end shaking at 20 rpm, samples were centrifuged at 4500 g for 15 min. The P content in the filtered supernatant was quantified by inductively coupled plasma-optical emission spectroscopy (ICP-OES) method.

It is noteworthy that all adsorption experiments were performed in triplicate and data were presented as the means of three repeats. Moreover, all used P compounds (myo-inositol hexakisphosphate: C$_6$H$_{18}$O$_{24}$P$_6$; α-glycerol phosphate: C$_3$H$_9$O$_6$P; potassium dihydrogen phosphate: KH$_2$PO$_4$) in the present experiments were of analytical grade chemicals and purchased from Carl Roth GmbH and Sigma-Aldrich. Working solutions were prepared fresh daily by adding accurate quantities of the prepared P stocks into 0.01 M CaCl$_2$ solution.

To describe the adsorption behaviors of P compounds on goethite, the adsorption data were fitted to the Freundlich [45], Langmuir [46], and Temkin [47] models. The Freundlich model provides a two-parameter equation that describes the relationship between the equilibrium concentration and the adsorbed one onto heterogeneous surfaces through the following equation:

$$Q_{ads} = K_f C_e^{n_f} \qquad (1)$$

that is rewritten in a linear form as follows:

$$\ln Q_{ads} = \ln K_f + n_f \ln C_e \qquad (2)$$

where $Q_{ads}$ is the amount of adsorbate adsorbed per unit mass (or surface area) of adsorbent, $C_e$ is the adsorbate equilibrium concentration in solution, $K_f$ is Freundlich adsorption

constant (sometimes it is defined as Freundlich unit adsorption capacity), and $n_f$ is the non-linearity exponent (Freundlich exponent) [45]. In the present study, $Q_{ads}$, $C_e$, and $K_f$ are expressed in µmol m$^{-2}$, µmol L$^{-1}$ and µmol$^{1-n_f}$ L$^{n_f}$ m$^{-2}$, respectively. Notice that the Freundlich model assumes that the adsorption enthalpy depends on the amount of adsorbate. In the limit of small $Q_{ads}$ where the adsorption enthalpy should not depend on $Q_{ads}$ one could describe the isotherm by a Langmuir model as well. The Langmuir adsorption theory assumes that the adsorbate forms a monolayer on a homogeneous adsorbent surface. The following equation expresses the Langmuir isotherm:

$$Q_{ads} = Q_{max} \frac{K_l C_e}{1 + K_l C_e} \tag{3}$$

that is rewritten in a linear form as follows:

$$\frac{C_e}{Q_{ads}} = \frac{C_e}{Q_{max}} + \frac{1}{K_l Q_{max}} \tag{4}$$

where $Q_{max}$ is the maximum amount of the adsorbate which is required to form a monolayer by complete saturation of all binding sites (it is defined also as the maximum monolayer coverage capacity or simply as monolayer capacity [5], µmol m$^{-2}$), $K_l$ is the Langmuir adsorption constant (L µmol$^{-1}$), which is mainly related to the adsorption energy [46]. The Temkin model exhibits a factor considering the adsorbent–adsorbate interaction with a uniform distribution of binding energies. The model is expressed by the following equation:

$$Q_{ads} = \frac{RT}{b_T} \ln A_T + \frac{RT}{b_T} \ln C_e = B_T \ln A_T + B_T \ln C_e \tag{5}$$

where $R$ is the universal gas constant (8.314 J K mol$^{-1}$), $T$ is the absolute temperature (K), $b_T$ is Temkin isotherm constant (J mol$^{-1}$), $A_T$ is Temkin isotherm equilibrium binding constant (L µmol$^{-1}$), and $B_T$ is constant related to the heat of adsorption. It is important to mention that our adsorption isotherm data were fitted by two ways. The first way is based on the normal linearization technique while the second way is based on non-linear equations solver function that was described by Bolster and Hornberger [48]. Fitted parameters obtained by both ways were compared to each other and the best fit parameters were selected based on error measures. Non-linear equations solver was found to be more appropriate compared to the linearization technique, and thus the corresponding parameters were selected and presented in the present contribution. For better understanding for the P adsorption behavior, the calculated adsorption coefficients are interpreted on three different bases (per mol P (i.e., per mole of P element), per mol molecule (i.e., per mole of P-containing molecule (OP, GP, and IHP)), and per mass (i.e., per mass (mg) of P-containing molecule)).

### 2.2. Molecular Modeling and Computational Details

The current molecular modeling approach for the binding process of phosphates at the goethite–water interface is illustrated in Figure 1. Here, three different abundant phosphates are considered, i.e., OP ($H_3PO_4$), GP ($C_3H_9O_6P$), and IHP ($C_6H_{18}O_{24}P_6$), to study the complexation reaction of each phosphate with the goethite surface in the presence of water (see Figure 1a–c). Each constructed goethite–phosphate–water complex model (for example, see Figure 1d), consists of 1– the goethite surface, 2– a phosphate compound with a specific binding motif to the surface, and 3– water molecules surrounding the goethite–phosphate complex. Here, the 100 goethite surface is considered, which is one of the most abundant goethite surface planes in soil systems [33,49]. This goethite surface plane is modeled by the repetition of the goethite unit cell (lattice constants: $a = 9.95$, $b = 3.01$, $c = 4.62$ Å) as $1a \times 8b \times 5c$ which consists of 640 atoms (160 Fe, 160 H, and 320 O atoms). The binding motifs formed between these phosphates and the goethite surface are modeled based on previous literature studies [9–11,41,50–52]. The initial motifs of OP and GP with the

goethite surface include monodentate (**M**, 1Fe + 1O one covalent bond between phosphate non-protonated O atom and a surface Fe atom) and bidentate (**B**, 2Fe + 2O i.e., two covalent bonds between two phosphate O atoms, one protonated and other non-protonated and two surface Fe atoms); see Figure 1e,f. For IHP, an additional, **4M** (4Fe + 4O i.e., four covalent bonds between four separate phosphates non-protonated oxygens and four surface Fe atoms) are considered since IHP is known to interact with goethite through multiple phosphate groups, see Figure 1g. To simulate the effect of water on the goethite–OP/GP/IHP complexes, each modeled binding motif is solvated with water molecules at a density of $\approx 1\,\mathrm{g\,cm^{-3}}$ perpendicular to the studied surface plane with height of ≈18 Å by using the visual molecular dynamics (VMD) package (see Figure 1d) [53]. To ease the discussion about the interactions, oxygen atoms bonded to surface Fe atoms are denoted as $O_p$.

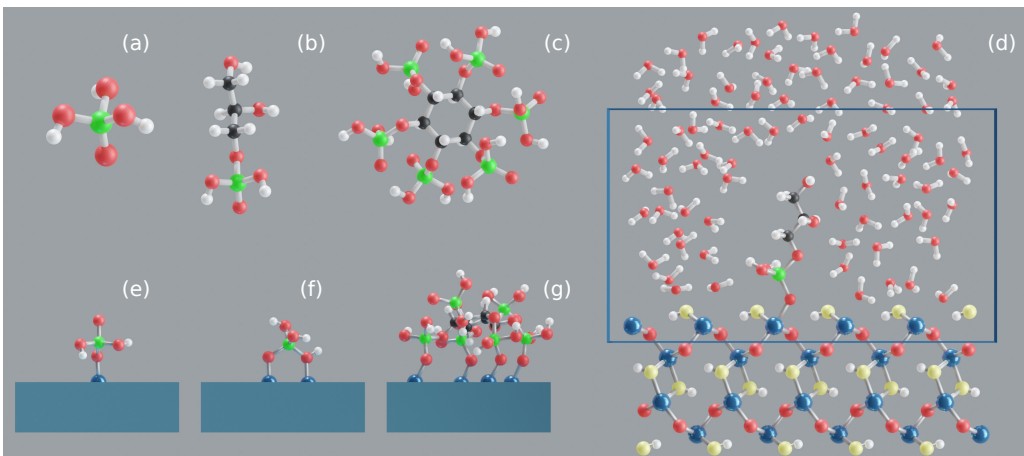

**Figure 1.** OP (**a**), GP (**b**), IHP (**c**), solvated goethite–GP complex highlighting QM box (**d**), goethite–OP **M** motif (**e**) and **B** motif (**f**), goethite–IHP **4M** motif (**g**). Blue, red, yellow, white, black and green colors correspond to iron, oxygen, hydroxyl oxygen, hydrogen, carbon and phosphorus atoms, respectively.

Due to the large size for the modeled goethite–OP/GP/IHP–water complexes (each model consists of more than 1200 atoms), the simulation of these systems with pure ab initio methods is computationally expensive and thus not feasible [54,55]. Therefore, the present models will be studied by molecular dynamics (MD) simulations based on the electrostatic embedding QM/MM hybrid approach. It should be noted that the present QM/MM approach has been validated for studying the P binding process at mineral/water interfaces. Specifically, this was achieved in two recent studies by comparing outcome of the QM/MM approach with corresponding results from pure DFT calculations [11] as well as experimental IR spectra [52]. Here, the reactive part (system of interest) is described at a QM level of theory, while the remaining part is described using MM. The QM part includes 1– OP/GP/IHP, 2– the goethite top surface layer (160 atoms), and 3– water molecules within a layer of about 10 perpendicular to the surface (≈55 molecules). The total QM box size is $22 \times 8b \times 5c$, i.e., $22 \times 24.08 \times 23.1$ Å. The QM part is simulated with density functional theory (DFT) as implemented in the quickstep code in CP2K [56]. A hybrid Gaussian and plane wave (GPW) dual basis method is used with ionic cores described by Goedecker-Teter-Hutter (GTH) pseudopotentials [57] in combination with the Perdew-Burke-Ernzerhof (PBE) [58] exchange correlation functional and the Grimme D3 empirical dispersion correction [59]. The valence electrons of all atoms are defined with the double-$\zeta$ valence polarized MOLOPT (DZVP-MOLOPT-SR-GTH) basis set except water for which the single-$\zeta$ valence (SZV-MOLOPT-SR-GTH) basis set is used to reduce computational cost [60]. The SCF convergence threshold was chosen to be $10^{-4}$ hartree. The MM part is described by classical force fields (FF) with FIST module in CP2K [61]. The goethite surface is modeled with the CLAYFF FF [62] while water is modeled with the single point charge (SPC) water model [63] and OP/GP/IHP with CHARMM FF using the SwissParm

tool [64]. Both CLAYFF and CHARMM FFs are compatible with the SPC water model. The interaction between the QM and MM parts in CP2K is implemented using the Gaussian expansion of the electrostatic potential method (GEEP) [65], wherein the MM charge is distributed by defining it using Gaussians instead of point charges to avoid electron spilling. QM/MM-based canonical (NVT, i.e., constant number of atoms N, volume V and temperature T) MD simulations are performed for 25 ps with a 0.5 fs time step while the temperature was maintained at 300 K using velocity rescaling thermostat (CSVR) [66]. The first 10 ps of each trajectory are assigned for the equilibration, and the remaining 15 ps (production trajectory) are used for analysis. The interaction energy between each phosphate and the goethite surface is calculated for every 100 fs (i.e., 150 snapshots) along the production trajectory by using:

$$E_{int} = E_{G-P\,complex} - (E_G + E_P) \qquad (6)$$

where, $E_{G-P\,complex}$, $E_P$, and $E_G$ denote the total electronic energies of the goethite–phosphate complex, the phosphate (OP/GP/IHP), and the goethite surface, respectively. The interaction energies involving water are divided by the number of water molecules in the simulation box for better comparison. The basis set superposition error (BSSE) in interaction energies is corrected using counterpoise scheme [67].

## 3. Results and Discussion

### 3.1. Adsorption Isotherms

Despite some data scatter in OP adsorption, Figure 2 shows that almost the maximum adsorption capacities are reached for goethite and the selected range of P concentrations is sufficient to achieve the equilibrium. Here, the sequence of adsorbed P ($Q_{ads}$) values per mol P is IHP > OP > GP (see Figure 2). For most cases, the coefficients of determination ($R^2$) values indicated that the Freundlich equation fitted the adsorption data better than the Langmuir and Temkin models; see Table 1. This comes in accord with the study by Tellinghuisen and Bolster [68] that discussed the statistical reasons leading to better fitting for the Freundlich equation case compared to Langmuir and Temkin models for the P adsorption on soils. The Freundlich adsorption constant (capacity) $K_f$ ranged from 0.22 to 4.79 $\mu mol^{1-n_f}\,L^{n_f}\,m^{-2}$. The order of $K_f$ values suggests that IHP exhibits the strongest adsorption and the highest capacity at the goethite surface, followed by OP and GP (GP < OP < IHP). The magnitude of the Freundlich exponent $n_f$, that ranged from 0.07 to 0.29 (see Table 1), gives an indication that the sorption mechanism is dominated by adsorption and not absorption [69,70]. Furthermore, the exponent points to the diversity of the energies associated with adsorption of P compounds on the goethite surface. Moreover, n < 1 for all cases indicates that upon increasing the P concentration/loading the binding energy between the surfaces and P compounds is reduced. The order of the $n_f$ values (IHP < GP < OP) suggests that the binding energy decreases with increasing the P loading in the order OP < GP < IHP. This means that affinity of the goethite surface to adsorb/bind a P molecule, with increasing the P concentration, increases in the order IHP < GP < OP. According to the Langmuir model, the maximum monolayer adsorption capacities ($Q_{max}$) are 1.2, 7.64 and 8.35 $\mu mol\,m^{-2}$, respectively, for GP, OP and IHP (see Table 1). This shows that the order of saturation of the goethite surface with P per mol P is GP < OP < IHP. This trend fits well with that one observed based on the Freundlich $K_f$ values. The Langmuir parameter $K_l$ increased in the order 0.001 (OP) < 0.003 (GP) < 0.06 (IHP) L $\mu mol^{-1}$. This constant is mainly related to the adsorption energy and could give information on how strong (i.e., strength) the goethite–P interaction/binding process is. Therefore, based on Langmuir model, one expects that the goethite–P interaction increases in the order OP < GP < IHP. The same order but with different values was obtained from Temkin binding constant $A_T$ which is also related to the binding strength, see Table 1. The Temkin $B_T$ values for OP (0.75), GP (0.13) and IHP (0.52) suggest that the heat of adsorption increases in the order GP < IHP < OP. Regardless the Temkin constant $B_T$, all other parameters from the

represented isotherm models in the present contribution refer to stronger adsorption and higher capacity for the IHP case compared to OP and GP cases by considering the number of P moles.

Considering the size and chemical structure of IHP (six phosphate groups), GP and OP (only one phosphate group), the adsorption data were presented as well in terms of whole P molecular system. Specifically, both $Q_{ads}$ and $C_e$ concentrations and all related fitted adsorption isotherm parameters are represented per mole and per mass of the whole P-containing molecule (OP, GP, and IHP), see Table 1. It should be noted that $n_f$ and $R^2$ values for all P compounds do not change upon changing the interpretation/representation of P concentration ($Q_{ads}$ and $C_e$). In addition, the order of all adsorption parameters ($k_f$, $n_f$, $k_l$, $Q_{max}$, $B_T$, $A_T$) does not change for OP and GP by different interpretations for the P concentration (i.e., per mol P versus per mol molecule and per mass bases). Here, the only change was observed for the IHP case. By considering the number of moles of the P adsorbed molecules (i.e., per mol molecule), the $K_f$, $Q_{max}$ and $B_T$ values for IHP decreased by a factor of six, while the corresponding $K_l$ and $A_T$ values increased by a factor of six (see Table 1). In principle, this means that the goethite adsorption capacity will decrease to one sixth by considering the whole IHP molecule compared to considering six P moles of each IHP mole. Consequently, the binding/adsorption strength between the goethite surface and each IHP mole will increase by a factor of six compared to the case of considering only one mole of P. In addition, the orders of $K_f$ (GP < OP < IHP), $K_l$ (OP < GP < IHP) and $A_T$ (OP < GP < IHP) do not change compared to the mol P basis. In contrast, the orders of $Q_{max}$ (GP < IHP < OP) and $B_T$ (IHP < GP < OP) changed comparing with considering the mol P cases. Same orders were obtained by considering the P concentration as mass of the adsorbed P molecule. With respect to the $K_l$ order, one can suggest that GP has stronger interaction than OP with the goethite surface. In contrast, the $B_T$ order suggests that OP has stronger interaction than IHP.

In a study evaluating the adsorption properties of goethite, Celi et al. [71] reported the adsorption ratio of 3:2 between IHP and OP at pH 4.5. Later, Martin et al. [72] confirmed the greater affinity of IHP than of OP for goethite, indicating that the maximum amount of adsorbed P were 3.6 and 2.4 $\mu$mol P m$^{-2}$ for IHP and OP, respectively. Likewise, Celi and Barberis [73] found that goethite shows a higher affinity for IHP (deduced from Langmuir $K_l$ values) than for other organic and inorganic phosphates. Li et al. [74] reported a maximum GP adsorption of 1.95 $\mu$mol P m$^{-2}$ at pH 5 which is close to the value of 1.20 $\mu$mol P m$^{-2}$ observed here. However, the lower adsorption of GP as compared to that reported by Li et al. [74] may be due to the differences in experimental conditions. Specifically, Li et al. [74] conducted their adsorption experiment by $\beta$-glycerophosphate at a goethite sample with SAA of 46.4 m$^2$ g$^{-1}$ and PZC of 9.2 in 0.1 M KCl as a background electrolyte. Overall, both organic and inorganic phosphates exhibited affinity for goethite. Parfitt [29], compared the adsorption of OP on goethite in CaCl$_2$ solution, and reported that larger amounts of specifically adsorbed P reflect more reactive and available adsorption sites on the goethite. This was explained by thermal gravimetric analyses that indicated that goethite contains 24% structural OH [74]. Similarly, Guzman et al. [75] showed the affinity for phosphate (exponent coefficient of Freundlich equation) for goethite on a surface area basis. Details of IHP adsorption on goethite have been extensively studied [71,72,76,77]. Ognalaga et al. [50] investigated the adsorption of IHP on synthetic goethite and attributed it solely to the inner-sphere complexation of phosphate groups with reactive OH groups. Celi et al. [71] and Ognalaga et al. [50] suggested that four of the six phosphate groups of each IHP molecule were involved in bonding to goethite while the other two remained free.

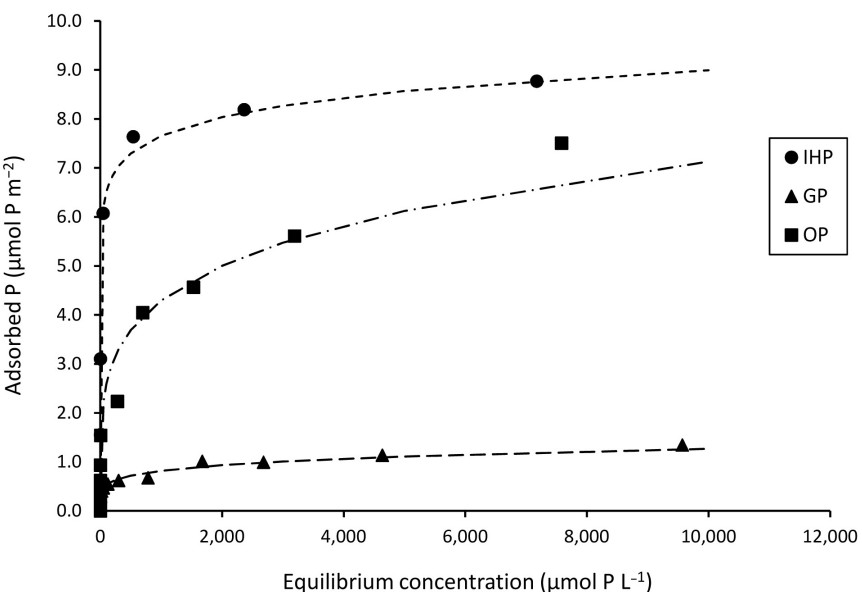

**Figure 2.** Adsorption isotherms of OP, GP, and IHP on goethite at 0.01 M CaCl$_2$ electrolyte solution (pH 5). Points represent the original data and dashed lines represent the fitted isotherms according to Freundlich model.

**Table 1.** Freundlich, Langmuir and Temkin isotherm coefficients ($K_f$, $n_f$, $K_l$, $A_T$ and $B_T$) of OP, GP and IHP adsorption on goethite at pH 5 in 0.01 M CaCl$_2$.

| Sorbate | Freundlich | | | Langmuir | | | Temkin | | |
|---|---|---|---|---|---|---|---|---|---|
| | $n_f$ | $K_f$ | $R^2$ | $Q_{max}$ | $K_l$ | $R^2$ | $B$ | $A_T$ | $R^2$ |
| OP (µmol P) | 0.29 ± 0.08 | 0.57 ± 0.23 | 0.97 | 7.64 ± 2.36 | 0.001 ± 0.001 | 0.88 | 0.75 ± 0.16 | 0.49 ± 0.23 | 0.83 |
| GP (µmol P) | 0.19 ± 0.01 | 0.22 ± 0.02 | 0.98 | 1.20 ± 0.06 | 0.003 ± 0.001 | 0.81 | 0.13 ± 0.03 | 0.89 ± 0.30 | 0.92 |
| IHP (µmol P) | 0.07 ± 0.001 | 4.79 ± 0.02 | 0.98 | 8.35 ± 0.04 | 0.06 ± 0.002 | 0.92 | 0.52 ± 0.01 | 3104.43 ± 467.8 | 0.99 |
| OP (µmol) | 0.29 ± 0.08 | 0.57 ± 0.23 | 0.97 | 7.64 ± 2.36 | 0.001 ± 0.001 | 0.88 | 0.75 ± 0.16 | 0.49 ± 0.23 | 0.83 |
| GP (µmol) | 0.19 ± 0.01 | 0.22 ± 0.02 | 0.98 | 1.20 ± 0.06 | 0.003 ± 0.001 | 0.81 | 0.13 ± 0.03 | 0.89 ± 0.30 | 0.92 |
| IHP (µmol) | 0.07 ± 0.001 | 0.90 ± 0.002 | 0.98 | 1.39 ± 0.006 | 0.36 ± 0.0009 | 0.92 | 0.09 ± 0.0004 | 18,444.8 ± 148.6 | 0.99 |
| OP (mg) | 0.29 ± 0.07 | 0.14 ± 0.06 | 0.97 | 1.04 ± 0.4 | 0.01 ± 0.01 | 0.88 | 0.10 ± 0.02 | 3.6 ± 1.1 | 0.83 |
| GP (mg) | 0.20 ± 0.01 | 0.05 ± 0.003 | 0.98 | 0.21 ± 0.01 | 0.02 ± 0.004 | 0.81 | 0.02 ± 0.001 | 5.19 ± 1.53 | 0.92 |
| IHP (mg) | 0.07 ± 0.001 | 0.61 ± 0.001 | 0.98 | 0.92 ± 0.004 | 0.55 ± 0.01 | 0.92 | 0.06 ± 0.0002 | 7427.1 ± 112.6 | 0.99 |

*3.2. Molecular Modeling of P Interactions at the Goethite–Water Interface*

3.2.1. Orthophosphate

The MD simulations showed that OP maintained the initial motifs (see Figure 1e,f) and formed stable **M** and **B** motifs throughout the simulation trajectory, see Figure 3a–h, respectively. The time averaged Fe–O$_p$ bond length and Fe–P distance observed for the **B** motif are shorter than those for the **M** motif which is probably due to higher stability in the former case, see Table 2. Here, the total interaction energy between OP and the goethite surface of the goethite–OP complex for the **B** and **M** motifs are –82 and –35 kcal mol$^{-1}$, respectively. The interaction energy per bond of goethite–OP in the **B** motif being higher than in the **M** motif suggests that the alignment of OP with the goethite surface favors the formation of an additional Fe–O$_p$ covalent bond in the **B** motif.

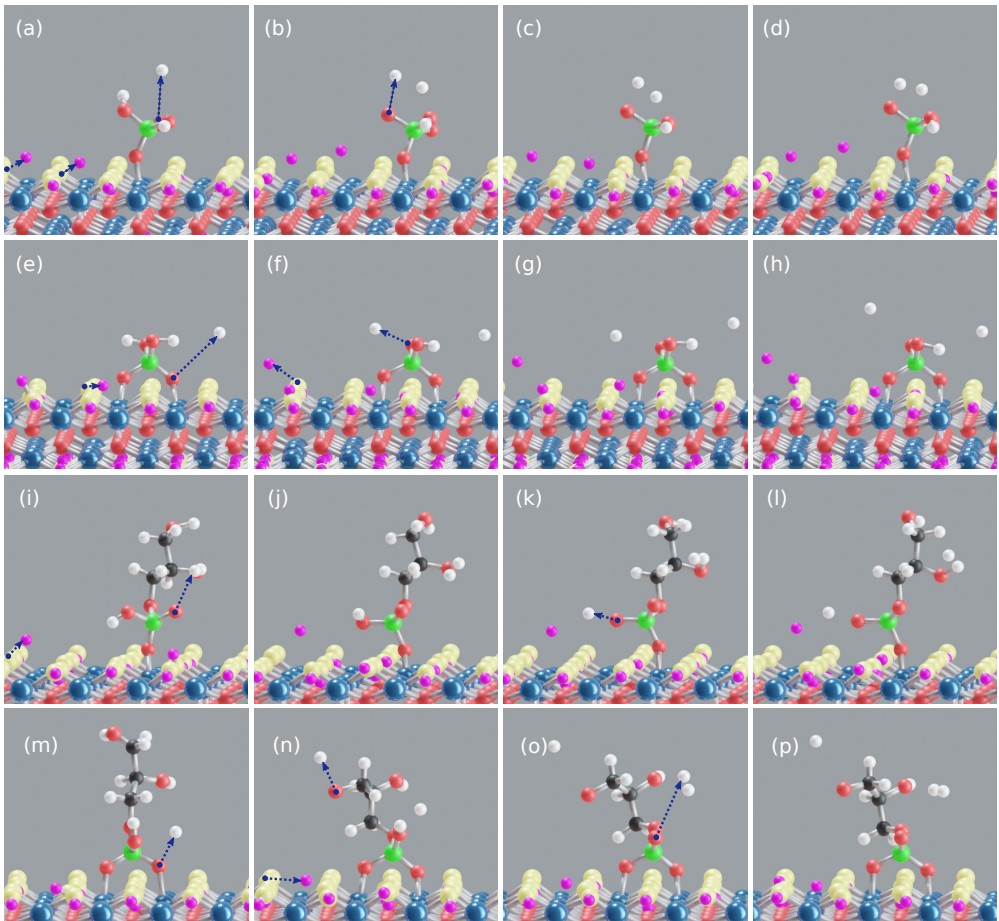

**Figure 3.** Snapshots along the simulation trajectories of the goethite–OP/GP/IHP–water complexes. OP **M** motif (**a**–**d**), OP **B** motif (**e**–**h**), GP **M** motif (**i**–**l**), GP **B** motif (**m**–**p**). Arrows denote proton transfers from phosphates to water and from goethite to water. For all cases, the first two snapshots are from equilibration phase while the last two are from production, except for OP **M** motif case where only (**a**) is from equilibration phase. The last snapshot for each case is taken at 25 ps. The goethite's hydrogen atoms and OP/GP/IHP's hydrogen atoms are shown in violet and white colors, respectively, for clear visualization of proton transfer events. Blue, red, yellow, black and green colors correspond to iron, bridging oxygen, hydroxyl oxygen, carbon and phosphorus atoms, respectively. The surrounding water is ignored here for better visualization.

**Table 2.** The interaction energies per number of bonds ($E_{int}$/bond) and selected interatomic distances Fe–$O_p$ and Fe–P observed along the trajectories of goethite–OP/GP/IHP–water complexes.

| P | Motif | $E_{int}$/bond (kcal mol$^{-1}$) | Fe–$O_p$ (Å) | Fe–P (Å) |
|---|---|---|---|---|
| OP | **M** | −35 | 2.1 | 3.5 |
| | **B** | −41 | 2.01 & 1.97 | 3.19 & 3.2 |
| GP | **M** | −48 | 1.99 | 3.19 |
| | **B** | −38 | 2.01 & 2.03 | 3.19 & 3.1 |
| IHP | **M(1)** | −85 | 2.0 | 3.1 |
| | **M(2)** | −69 | 2.03 | 3.2 |
| | **3M** | −72 | 2.01 & 1.98 & 1.94 | 3.1 & 3.2 & 3.4 |

For both **M** and **B** motifs, two protons are transferred from OP to water (see Figure 3a,b,e,f), forming an average of five HBs each. Consequently, the calculated OP–water interaction energy per water molecule for **M** (−1.7 kcal mol$^{-1}$) and **B** (−1.6 kcal mol$^{-1}$) motifs is essentially the same. In addition, proton transfers are observed from goethite to water, see Figure 3a,e,f. The proton transfers observed here occurred only during the

equilibration phase of the complexes and OP remained twice deprotonated in the production trajectory for both cases. The goethite–water interaction energy per single water molecule ($-4.8$ kcal mol$^{-1}$) is lower than the goethite–OP interaction energy for both **M** and **B** motifs. The goethite–water interaction energy is higher than OP–water interaction energies in both motif cases because of goethite's larger solvent accessible surface area (SASA), proton transfer events from goethite to water (see Figure 3), formation of multiple Fe–O$_{H_2O}$ **M** motifs (on average 15 of 40 surface Fe atoms, see Figure S6) and HBs between goethite and water.

### 3.2.2. Glycerolphosphate

GP is aligned perpendicular to the goethite surface to form initial **M** and **B** motifs, as shown in Figures 1d and S3c,d. The MD simulations show that these initial motifs are stable during the whole trajectory (see Figure 3i–p, respectively). The time averaged Fe–O$_p$ bond length and Fe–P distance observed for the **M** motif are close to the corresponding values for the **B** motif case; see Table 2. GP forms an average of six and eight HBs with water in the **M** and **B** motifs, respectively. It also exhibits multiple proton transfers to water, two in the **M** motif case and three in the **B** motif. The proton transfer events observed for both cases occurred during the equilibration phase. The total interaction energy between GP and goethite for the **B** motif ($-76$ kcal mol$^{-1}$) is higher than for the **M** motif ($-48$ kcal mol$^{-1}$) due to additional covalent bond in former motif. The higher overall goethite–GP interaction energy of the **B** motif indicates that it is a more favorable motif than the **M** one. However, the goethite–GP interaction energy per bond for the **B** motif is lower than for the **M** motif, suggesting comparatively unfavorable conditions for Fe–O$_p$ bonds in former motif. This is in contrast to the case of OP. The difference comes from the glycerol group in GP. Both OP and GP interact through the phosphate, but through the glycerol group, GP has a higher SASA and exhibits stronger interaction with water than OP (OP/GP–water interaction energies will be discussed below). In the GP **B** motif case, two oxygens of GP's phosphate group are bound to two individual Fe atoms and when the glycerol group interacts with flexible water atoms, GP fluctuates with water while the surface Fe atoms competitively pull O$_p$ oxygens towards the surface as a counteraction. This may induce strain in the Fe–O$_p$ bonds for the **B** motif and, therefore, the goethite–GP interaction energy per bond for the **B** motif is less compared to the **M** motif. The stretch in the Fe–O$_p$ bonds found in the GP **B** motif is larger than in the OP **B** motif, see Figure S4, which signifies a strain in the Fe–O$_p$ bonds in the former case. In addition, GP is found to oscillate occasionally in a seesaw type of motion over the surface which randomly stretches one of the Fe–O$_p$ bonds to its extreme (maximum Fe–O$_p$ bond length found $\approx$2.4 Å), see Figure S5. A similar observation was made for GP on 100 diaspore surface [41] which is isomorphous with goethite [33]. The study by Xu et al. [78] showed that GP induces more negative charges onto hematite mineral surface than OP, which might also influence the strength of the Fe–O$_p$ covalent bonds.

Regarding GP–water interaction energies, GP exhibited a slightly stronger interaction with water for the **B** motif ($-2.8$ kcal mol$^{-1}$) case than for the **M** motif one ($-2.5$ kcal mol$^{-1}$). This is due to additional proton transfer observed from GP to water for the **B** motif case (see Figure 3m–o). The GP–water interaction energy here is higher than that of OP–water due to additional glycerol group in GP which contains two polar OH groups that interact strongly with water. Similar to goethite–OP–water complexes, proton transfer is observed from goethite to water in both motifs here; see Figure 3i,n.

### 3.2.3. Inositolhexaphosphate

IHP was initially aligned perpendicular to the goethite surface to form **M** and **B** motifs (see Figure S3a,b), and parallel to the goethite surface to form the proposed **4M** motif (see Figure 1g). In contrast to goethite–OP–water and goethite–GP–water cases, transformation of initial binding motifs between IHP and the goethite surface with time is observed. The simulation results show that the initial **M** and **B** motifs resulted in two similar con-

formers of **M** motif type; namely, **M(1)** and **M(2)**. On the other hand, the starting **4M** binding motif transformed along the MD trajectory into a **3M** motif. The transformations observed here happened within the equilibration phase of each complex. The corresponding Fe–$O_p$ bond lengths and Fe–P distances of the relatively long-lived binding motifs [**M(1)**, **M(2)**, **3M**] are in close range and lie within half angstrom difference (see Table 2). The **M(1)** motif's goethite–IHP interaction energy is higher than that for the **M(2)** one (see Table 2). This is probably because of the strain in the Fe–$O_p$ bond in the **M(2)** motif due to intramolecular HBs between the phosphate group bound to Fe and its adjacent phosphate groups (see Figure 4h). The **3M** motif has a higher total interaction energy than both **M** motifs due to additional covalent bonds and proton transfer from IHP to the goethite surface. However, the interaction energy per bond between IHP and goethite for the **3M** binding motif is less than for the **M(1)** motif. Thus, in spite of the **3M** motif being the more favorable motif compared to the **M** motifs, the alignment of IHP with goethite as in the **3M** motif weakens the overall strength of the individual Fe–$O_p$ bonds.

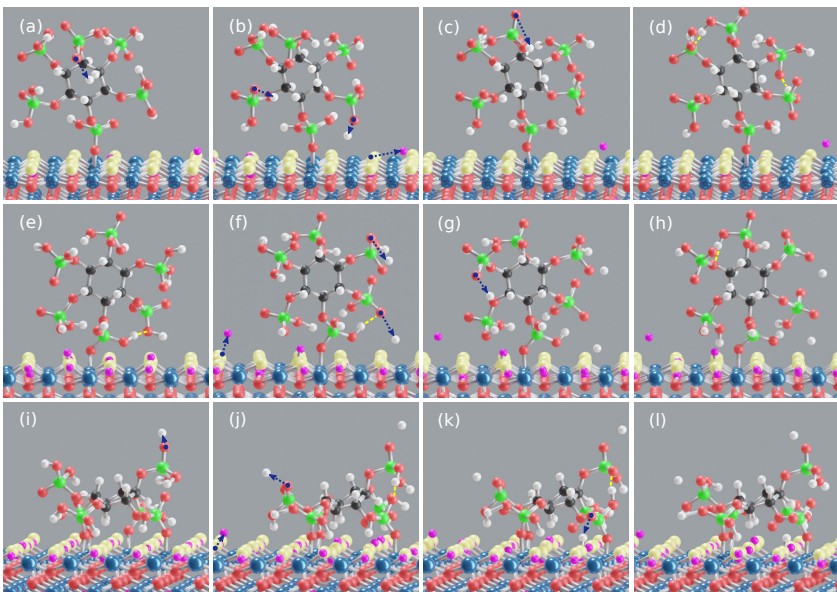

**Figure 4.** Snapshots along the simulation trajectories of the goethite–IHP–water complexes. IHP **M** motif (**a–d**), IHP **B** motif (**e–h**), IHP **3M** motif (**i–l**). Arrows denote proton transfers from phosphates to water and from goethite to water. The yellow dotted lines denote intramolecular HBs between IHP's phosphate groups. For all cases, the first two snapshots are from equilibration phase while the last two are from production, except for the IHP **3M** motif case where only (**i**) is from the equilibration phase. The last snapshot for each case is taken at 25 ps. The goethite's hydrogen atoms and IHP's hydrogen atoms are shown in violet and white colors, respectively, for clear visualization of proton transfer events. Blue, red, yellow, black and green colors correspond to iron, oxygen, hydroxyl oxygen, carbon and phosphorus atoms, respectively. The surrounding water is ignored here for better visualization.

Even though an additional proton transfer is observed from IHP to water for the **M(1)** motif, the IHP–water interaction energy ($-7.3$ kcal mol$^{-1}$) is very close to that for the **M(2)** motif case ($-7.2$ kcal mol$^{-1}$). This could be probably due to higher number of time averaged HBs observed for the **M(2)** motif case (25) compared to the **M(1)** case (23). The IHP–water interaction energy observed for the **3M** motif case ($-3.1$ kcal mol$^{-1}$) is lower than those for both **M** cases due to fewer proton transfer processes (2) and HBs (15) observed in the **3M** motif case.

The stronger interaction of IHP with water is vital for understanding the reason for transformation of initial **B** and **4M** motifs to **M(2)** and **3M** final motifs, respectively. For initial **B** motif case, when IHP interacts with flexible water molecules it fluctuates with respect to goethite surface which might induce strain in one of the Fe–$O_p$ covalent bonds

leading to its dissociation. In addition, an intramolecular HB is observed between the Fe bonded phosphate group and its adjacent phosphate group, see Figure 4l. In our previous work on IHP interaction at the diaspore(100)–water interface (diaspore isostructural with goethite), [41] the initial **4M** motif was transformed into a **2M** motif. The reason is that the Al–$O_p$ covalent bonds are inclined and restricted between diaspore's surface hydroxyl groups. Moreover, when IHP interacts with water and fluctuates during the equilibration phase, the restricted Al–$O_p$ covalent bonds under unfavorable conditions do not have an alternative but to dissociate and move towards water. The same reason holds true here for the transformation process of the initial **4M** motif to a stable **3M** binding motif.

*3.3. Discussion of Experimental and Modeling Results*

In what follows, we will correlate the present modeling results and the adsorption isotherms as per mole of P-compound. It is important to emphasize that the experimental adsorption results are due to a combination of all different binding motifs at the existing different surface planes (not only the 100 goethite surface plane) in the real goethite sample. Therefore, a direct quantitative comparison is rather difficult and thus we are aiming here to present a qualitative behavior. First, let us recollect that $K_f$, $K_l$ and $B_T$ relate to the phosphate's binding energy and that $n_f$ relates to the binding energy of the next incoming P molecule to surface (i.e., binding affinity). Since the Freundlich isotherm provided the best fit to the experimental data for all cases (see Table 1), we will discuss the theoretical results in Table 2 in the context of $K_f$ and $n_f$. Analyzing the strength of phosphate interaction shows that the $K_f$ values are in line with the order of overall binding energies observed here, i.e., GP **B** < OP **B** < IHP **3M**. The $K_f$ value for OP adsorption is more than twice that of the GP adsorption, suggesting a significant difference between OP and GP interaction with goethite. In light of these results, combined with the theoretical binding energy values (see Table 2), one could suggest that OP might predominantly form **B** motif and GP might form **M** motif as the predominant binding motif. Therefore, the binding strength order could be updated as GP **M** < OP **B** < IHP **3M**. Comparing $K_f$ values for IHP and OP, the $K_f$ for IHP adsorption is about 1.58 times that for OP adsorption, which is very close to the reported adsorption ratio by Celi et al. [71] and Martin et al. [72]. Correlating this ratio to the present calculated binding energies would suggest that one should rather use the average of all three bonding motifs, i.e., both **M** and **3M** binding motifs for the adsorbed IHP molecule are present at the goethite surface. Therefore, based on both experimental and theoretical binding strength values, one could suggest that the overall binding strength increases in the order GP (**M**) < OP (**B**) < IHP (**M + 3M**).

The order of $n_f$ values (IHP < GP < OP, see Table 1) suggests that IHP adsorption leads to a faster saturation of the goethite surface than GP and OP. Even though IHP binds to goethite through a few phosphate groups, the remaining phosphate groups (often deprotonated) would induce more negative charges to the surface compared to OP and GP cases [71,78]. Moreover, it was found that GP leads to a more negatively charged surface than OP [78]. This explains the reason why the $n_f$ value for GP is lower than for OP.

Let us compare the present modeling results with previous experimental and theoretical studies. In the case of OP, experimental [36,79,80] and theoretical [10,40] results pointed to a formation of both, i.e., **M** and **B**, motifs with goethite. Specifically, for the 100 goethite surface studied here, Kubicki et al. [40] and Ahmed et al. [10] demonstrated that OP often exists in a doubly deprotonated state forming both **M** and **B** binding motifs but with the predominance of the **B** one. This is confirmed by the present MD results, i.e., OP has formed stable **M** and **B** motifs and remained twice as deprotonated in the production trajectory with the **B** motif exhibiting higher interaction energy with the goethite surface compared to the **M** motif. Even though the **B** motif is the more stable motif under the present conditions, it is not always the dominant one, a topic which has been highly debated in the literature [39,40,79,80]. For instance, Persson et al. [80] interpreted their FTIR spectra of the OP adsorption at goethite and hematite by the formation of only **M** binding motifs with different protonation states. This analysis was done as a function of total phosphate concen-

tration, pH, and time. In contrast, Tejedor-Tejedor and Anderson [36] proposed that the **B** motif is the dominant motif, for low surface coverage of OP on goethite over the pH range of 3.6–8.0. Further, they proposed that the **M** motif exists at low surface coverage but as a non-dominant motif. By using DFT calculations, Kwon and Kubicki [39] concluded that the **B** motif is dominant between pH 4–6. Recently, a joint approach by Ahmed et al. [51] involving adsorption experiments and DFT simulations showed that the **M** motif is dominant at both extremely low and high pH values, while the **B** motif is dominant in the intermediate pH range.

Keeping in mind that the present simulations correspond to a low surface coverage scenario and acidic pH, one can conclude that the stable **M** and **B** motifs with abundance of the **B** one here are in line with the studies of Tejedor-Tejedor and Anderson [36], Ahmed et al. [51], and Abdala et al. [79]. Moreover, the time averaged Fe–P distance observed in both motifs (3.5 Å for the **M** motif and 3.19–3.2 Å for the **B** motif) are within the range of values reported in literature, thus giving further support for the present model. For instance, an extended X-ray absorption fine structure study of adsorbed OP on goethite [79] showed that the distances between the surface Fe atoms and the adsorbed phosphate P atoms (i.e., Fe–P distances) for **M** and **B** binding motifs are 3.6 and 3.28 Å, respectively. The Fe–P distances from other studies are in the range of 3.48–3.55 Å for the **M** motif and 3.13–3.37 Å for the **B** motif [10,40,81,82].

Compared to OP, GP adsorption on goethite has not been extensively studied and hence, the information about its binding motifs with goethite is limited. Li et al. [74] showed that GP forms inner-sphere complexes with goethite by replacing water and hydroxyl groups from surface active sites. The study also proposes that GP forms only **M** motifs at the goethite surface based on FTIR spectra analysis. The **B** motif is sterically hindered by the organic moiety. Additionally, Hartree-Fock simulations and FTIR spectra studies by Persson et al. [83] of monomethyl phosphate ($CH_3–H_2PO_4$), an organic P with a single phosphate group like GP, showed that it forms mostly **M** motifs with goethite. Adsorption isotherms and FTIR spectra by Sheals et al. [34] proposed that glyphosate binds predominantly through an **M** motif but **B** motifs might be formed at low P concentration and neutral pH. By using periodic DFT based MD simulations, Ahmed et al. [9] studied the glyphosate binding process at the goethite–water interface by considering three goethite surface planes. The results indicated that the **M** binding motif is the most dominant one at the 100 goethite surface plane, although both **M** and **B** motifs exist at the 100, 010 and 001 surface planes. Summing up these literature data, the **M** motif is the dominant motif for GP like molecules at the 100 goethite surface. This could be supported by our recent study [52] about the P binding at the 010 goethite surface plane. We found that the calculated total interaction energy for the GP **B** motif ($-122$ kcal mol$^{-1}$) is higher than for the **M** motif ($-112$ kcal mol$^{-1}$). However, the goethite–GP interaction energy per bond for the **B** motif is lower than for the **M** motif. This suggests unfavorable conditions for Fe–O$_p$ bonds for the **B** case. This is in accord with outcome of the present contribution. In addition, the calculated interaction energy indicated that the 010 goethite surface plane binds GP stronger than the corresponding 100 surface plane. Regarding the Fe–P distances observed for both GP motifs at both 010 and 100 goethite surface planes, the distances are within the range of Fe–P distances observed for OP on goethite in the literature [10,40,79,81,82]. This suggests that the GP's phosphate group interacts with goethite surface in a similar way to the OP case.

There is no consensus in the literature about the number of IHP's phosphate groups that bind to goethite surface or about dominant binding motifs. The binding motifs observed for goethite–IHP complexes in the current study (**M** and **3M**) have been found in literature before [11,41]. Similarly, we have observed the same binding motifs (i.e., **M** and **3M**) for IHP at the 010 goethite surface plane [52]. Similar to the GP case, the calculated interaction energy referred to stronger interaction for IHP at the 010 goethite surface plane compared to the 100 surface plane. It is important to mention that it is not clear yet in literature which binding motif is the dominant one. For instance, **3M** [11,84], **2M** [11,41]

and **1M** [11] are different binding motifs observed for IHP on minerals. In contrast, Johnson et al. [77] proposed that IHP interacts with goethite by forming outer-sphere complexes. Interestingly, none of the above mentioned studies for IHP suggest that it forms a bidentate binuclear (**B**) motif. Similar to our previous studies [11,41], we found that the initial **B** motif of IHP was not stable due to the strong interaction of IHP with water and intramolecular HBs. Therefore, we propose that IHP phosphate groups form **M** motif with goethite surface.

De Groot and Golterman [85] showed that IHP adsorption onto goethite had an inhibitory effect on OP adsorption and it can release adsorbed OP from the goethite surface. The same could be inferred from binding energies here, where the goethite–IHP binding energies are larger than for goethite–OP complexes. The goethite–GP binding energies are also less than goethite–IHP and one might suggest that IHP might replace GP as well from goethite surface. The same could be inferred from the adsorption strengths order calculated from experiments here. The Fe–P distances observed here show that they are close to the Fe–P distances for OP on goethite [10,40,79,81,82] which suggests that IHP's organic moiety might not influence the individual phosphate groups interaction with goethite but only the conformational flexibility of the overall binding motif [11,41]. This comes in accord with the observation by Celi et al. [71] that phosphate groups of IHP react with a goethite surface similar to OP.

## 4. Summarizing Discussion

The current study contributes to the efforts in understanding the interaction of phosphates with soil minerals. Here, both experimental and theoretical approaches are adopted to characterize inorganic (OP) and organic (GP, IHP) phosphates interaction with abundant and reactive goethite mineral. The goethite–OP/GP/IHP–water complexes are simulated with the multiscale QM/MM method which provides molecular level insights into adsorption experiments performed for OP/GP/IHP on goethite. The binding energies and interaction mechanisms of phosphates adsorption on goethite from the modeling study are correlated to adsorption data fitted to the Freundlich isotherm, which provided a uniformly better fit than the Langmuir and Temkin models. The model coefficients provided an overview of the pattern of phosphate interactions with goethite and the QM/MM simulations demonstrate, at the molecular level, the attributes that build these pattern.

The modeling results show that OP forms stable **M** and **B** motifs and OP **B** motif has higher interaction energy per bond than OP **M** motif. This suggests that goethite–OP interaction favors the additional covalent bond, which makes the OP **B** motif more stable. For goethite–GP complexes, the order of interaction energies is same as OP case, except that GP **B** motif interaction energy per bond is lower than GP **M** motif. Compared to OP, the GP **B** motif is weaker than OP **B** motif due to a strain in Fe–$O_p$ bonds in former case. The strain is due to GP interaction with water molecules. In addition, Xu et al. [78] study shows that adsorption of GP induces more negative charges on the hematite surface than OP which might strain the Fe–$O_p$ bonds further. Therefore, the literature [34,74] and the calculated binding energies in the present contribution show that GP's dominant motif might be the **M** or **B** motif depending on the GP interaction with environmental molecules. IHP forms **M** and **3M** motifs with multiple intramolecular HBs between adjacent phosphate groups [77]. IHP's **3M** motif exhibits the strongest binding energy with the goethite surface among all goethite–OP/GP complexes here. This comes in accord with our recent study [52] for the binding of GP and IHP at the 010 goethite surface plane. In that study, the **3M** binding motif of IHP showed the strongest interaction with goethite compared to all other existing binding motifs for GP and IHP. Therefore, we propose that the **3M** motif might be the most dominant motif at low surface loading.

The adsorption data from experiment fitted to Freundlich model show that the order of adsorption strength ($K_f$) is GP < OP < IHP whereas the order of incoming P binding strength with surface ($n_f$) is IHP < GP < OP. Based on the magnitude and order of experimental $K_f$ values combined with theoretical binding strength values, one could suggest that the overall binding strength increases in the order GP (**M**) < OP (**B**) < IHP (**M**

**+ 3M**). This shows that GP might often form **M** motif, whereas OP might form **B**, and IHP might form both **M** and **3M** motifs. The $n_f$ order suggests that IHP adsorption on goethite saturates its surface faster than for the GP and OP cases. The fact that GP adsorption induces more negative charges on goethite than OP supports our suggestions that GP would often form **M** and not **B** motifs and eventually its interaction energy would be less than OP.

Eventually, the present experimental and theoretical results, in agreement with the pertinent literature, plausibly explain why IHP is the predominant organically bound P form in soil. A challenge for further studies and application is to understand and explore the mechanisms by which microbes and plants can overcome the strong IHP–mineral binding and incorporate the phosphate groups into their metabolism.

**Supplementary Materials:** The following are available online at https://www.mdpi.com/2075-163X/11/3/323/s1, Figure S1: XRD pattern for pure goethite sample, Figure S2: Effect of pH on goethite zeta potential, Figure S3: Initial motifs of IHP GP, Figure S4: Sum of covalent bond lengths in OP and GP **B** motifs, Figure S5: Seesaw movement of GP's phosphate group, Figure S6: Multiple covalent bonds between surface irons atoms and water molecules.

**Author Contributions:** P.B.G. and M.M. performed the modeling and experimental work, respectively, and wrote the manuscript draft. A.A.A., P.L. and O.K. are responsible for conceptualization, funding acquisition, methodology, supervision, writing–review and editing of the present contribution. All authors have read and agreed to the published version of the manuscript.

**Funding:** We are grateful to S. Dultz, Soil Science, University of Hannover, for preparing and characterizing the goethite. We gratefully acknowledge the financial support by the German Research Foundation (DFG) as a part of the SPP 1685 Priority program "Ecosystem Nutrition: Forest strategies for limited phosphorus resources" (P.B.G., O.K., A.A.A.) and the InnoSoilPhos-project (A.A.A.), funded by the German Federal Ministry of Education and Research (BMBF) in the frame of the BonaRes-program (No. 031A558). This research was performed within the scope of the Leibniz Science Campus "Phosphorus Research Rostock". The authors thank the North German Super computing Alliance for providing HPC resources (project mvp00016).

**Institutional Review Board Statement:** Not applicable.

**Informed Consent Statement:** Not applicable.

**Data Availability Statement:** Not applicable.

**Conflicts of Interest:** The authors declare no conflict of interest. The funders had no role in the design of the study; in the collection, analyses, or interpretation of data; in the writing of the manuscript, or in the decision to publish the results.

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
