# Peer review of "The Binding of Phosphorus Species at Goethite: A Joint Experimental and Theoretical Study"

_minerals, doi:10.3390/min11030323_

Round 1
Reviewer 1 Report
This is an interesting study with some novel approaches to the modeling. I have some questions I would like to see addressed before I can recommend publication.
1. The authors selected the (100) surface of goethite, I suppose because this is a common or predominant surface. However, they are aware of the work of Mario Villalobos and co-workers who have shown that faces only present in a minor amount can do most of the adsorption of given ions and compounds. The authors own work as cited states that the (010) surface of diaspore was a stronger face for GP and IHP, so why not include (010)?
2. The experimental goethite must have more than the (100) surface present. Did the authors characterize which surfaces were present and in what percentages?
3. Given that the experimental isotherms are the result of adsorption to all faces, how can a direct comparison of the observations and models of the (100) surface only be conducted?
4. The QM/MM approach is interesting, but what testing have the authors done to ensure its accuracy? One does not expect any modeling method to be 100% accurate, but it is worthwhile to have an estimate of the error in predicted parameters such as structures and energetics.
5. It is often helpful to have at least one type of spectroscopic measurement of the surface complexes in order to compare to the modeled structures. The authors discuss previous work on this, but it does not appear they attempted to reproduce any IR spectra with their models. This seems possible based on the method of extracting molecular clusters from the periodic models.
Author Response
Response to reviewer is attached.

Reviewer 2 Report
General
This paper seeks to understand differences in sorption mechanisms and strength for phosphate compounds: orthophosphate (OP), glycerolphosphate (GP) and inositolhexaphosphate (IHP), onto the 100 surface plane of goethite. They use traditional sorption experiments and fitting to Freundlich, Langmuir and Temkin models, followed by molecular dynamics (MD) simulations. They find different binding mechanisms – monodentate, bidentate, and (?) multiple monodentate (3M) mechanisms for the three compounds. They have excellent graphics from the MD simulations that demonstrate the binding mechanisms and they do a great job of summarizing existing literature. This paper is well written, very interesting and insightful, and will be of interest to Minerals readers. I really enjoyed the read. I have very few comments for improvement.
Specific comments:
L117 Want to make author aware of a paper that describes a problem with linear fitting to nonlinear equations, particular to sorption experiments. Basically, because the x and y axis are dependent on each other (& calculated from the same data), linearized equations may not be appropriate. Bolster and Hornberger 2006: Soil Sci. Soc. Am. J. 71:1796–1806. doi:10.2136/sssaj2006.0304
Table 1 should have standard deviations or standard errors of the fitted parameters
Fig 4: just want to say how much I love these figures! Could you add the times for each of the snapshots?
L351: why does GP lead to a more negative charged surface compared to OP?
Section 3.3: I think this literature review section is VERY effective.
L376: can you add more detail on the EXAFS results? What does your statement mean?
Section 4: this section is so clearly written! Very nice.
L432: very much enjoyed the ability to model the water molecules, very insightful. In some ways I don’t feel I have enough explanation of the MD simulations. They are sort of presented as fact, and not really as a results and then interpretation of them. Can you present them in a more conventional fashion as results and discussion?
In several places in the paper, it is stated that sorption of one compound is faster than another. From where does this information come? I didn’t see mention of kinetic experiments or see results. And the MD simulations only say the time for the last picture (25 ps). Is there data or more information that should be shown to support points about rates of sorption?
The authors note that the Frendlich equation makes the better fit. Want to make authors aware of another Bolster paper (full disclosure, I am not Bolster) that finds statistical reasons as to why Freundlich equation is better than Langmuir and Temkin models (on whole soils): Environ. Sci. Technol. 2010, 44, 5029–5034
I would be interested to hear about whether your results would be different if on a different goethite plane, can you add a few sentences on that?
Author Response
Response to reviewer is attached.

Reviewer 3 Report
Review on “The Binding of Phosphorus Species at Goethite: A Joint Experimental and Theoretical Study” by Ganta et al. (minerals-1119126)
The manuscript studies the binding of P species onto goethite. Specifically, the adsorption of common phosphates such as orthophosphate, glycerolphospate and inositolhexaphosphate. Both, adsorption experiments and hybrid ab initio molecular dynamics simulations were conducted to understand the mechanism of this adsorption process.
In this study, the authors synthesized goethite and added different concentrations of the P species in order to obtain adsorption isotherms. They used different empirical models to describe and predict the data behaviour, thus obtaining good fits. The authors also focused on the understanding of the molecular interaction between the P species and the goethite surface, obtaining good model predictions which are in line with previously published data.
It is to my understanding that the topic is relevant and fits well with the scope of the journal. I consider that it would be very interesting to focus on how these different species would interact with the surface of other minerals and in the presence of other competing species. However, I am aware that this falls out of the reach of the current study. It is my opinion that this manuscript can be considered for publication in Minerals after very minor corrections.
Specific comments:
- Mention the limitations of the Brunauer-Emmett-Teller method to measure the specific surface area.
- In the modelling part, mention and briefly compare with previous works in which other modelling approaches were followed.
- Even when the English is correct and the results are well explained, I would suggest a revision on the style to ease the reading (i.e. use of more paragraphs and shorter sentences).
Author Response
Response to reviewer is attached.

Round 2
Reviewer 1 Report
I am OK with the authors' response and revisions.